# Industrial Processing Affects Product Yield and Quality of Diced Tomato

Francesco De Sio [1], Mariateresa Rapacciuolo [1], Alessandro De Giorgi [1], Luca Sandei [1], Bonaventura Giuliano [2], Alessio Tallarita [3], Nadezhda Golubkina [4], Agnieszka Sekara [5,*], Vasile Stoleru [6,*], Antonio Cuciniello [7], Giuseppe Morano [3] and Gianluca Caruso [3]

1   Experimental Station for Food Preserving Industry of Parma, Angri Branch, 84012 Salerno, Italy; francesco.desio@ssica.it (F.D.S.); mariateresa.rapacciuolo@ssica.it (M.R.); alessandro.degiorgi@ssica.it (A.D.G.); luca.sandei@ssica.it (L.S.)
2   National Association of Food Preserving Manufacturers, 80143 Naples, Italy; b_giuliano@anicav.it
3   Department of Agricultural Sciences, University of Naples Federico II, Portici, 80055 Naples, Italy; lexvincentall@gmail.com (A.T.); giu.morano90@gmail.com (G.M.); gcaruso@unina.it (G.C.)
4   Federal Scientific Center of Vegetable Production, Selectsionnaya 14 Vniissok, 143072 Moscow, Odintsovo, Russia; segolubkina45@gmail.com
5   Department of Horticulture, Faculty of Biotechnology and Horticulture, University of Agriculture, 31-120 Krakow, Poland
6   Department of Horticultural Technologies, "Ion Ionescu de la Brad" University of Agricultural Sciences and Veterinary Medicine, 3 M. Sadoveanu, 700440 Iasi, Romania
7   Council for Agricultural Research and Economics (CREA)—Research Center for Cereal and Industrial Crops, 81100 Caserta, Italy; antonio.cuciniello@crea.gov.it
*   Correspondence: agnieszka.sekara@urk.edu.pl (A.S.); vstoleru@uaiasi.ro (V.S.)

**Abstract:** The tomato industry has been searching for new genotypes with improved fruit production, both in the field and industrially processed, together with high-quality performance under sustainable management conditions. This research was carried out in Southern Italy with the aim of assessing the effects of industrial processing on the yield and quality of four tomato hybrids grown according to organic farming methods and addressed at dicing. MAX 14111 and HMX 4228 showed the highest values of field and processing yield as well as reduced sugars and fructose. MAX 14111 had the highest values of total solids and soluble solids, titratable acidity, fiber, energetic value, polyphenols, and also rutin, though not significantly different from Impact. HMX 4228 performed best in terms of sugar ratio, color and naringenin. Concerning the diced products, the sensorial qualities of the four hybrids differed significantly. Total polyphenols, naringenin and rutin in the tomato fruits were higher in the processed than in the raw product. The appreciable fruit yield and quality resulting from both field and processing phase represent a promising perspective for identifying improved tomato genotypes addressed at dicing.

**Keywords:** *Solanum lycopersicum* L.; round-prismatic fruit genotypes; field and processed production; color; sugars; polyphenols; rutin; naringenin; lycopene

## 1. Introduction

Farmers are currently aware that crop systems should be managed achieving high food quality while safeguarding environmental resources. The organic farming method is deemed to meet the aforementioned demands, though it is supposed to give lower yields, therefore needing more land to produce the same amount of food as compared to conventional farming [1]. In the same way as vegetables directed to the fresh market, industrial crops can also be adequately managed in compliance with the organic farming procedures and, in the case of tomato, it is essential to identify hybrids providing a high production of fruits showing appreciable nutritional and organoleptic properties. Tomato (*Solanum Lycopersicum* L.) is one of the most representative industrial crops and accounts

for a global production of 182.3 million tons [2]. It is much appreciated due to its richness in health-promoting compounds such as minerals, carotenoids, vitamins, flavonoids, and polyphenols. The latter molecules represent a healthy resource for humans and a resistance factor to plant adversity, even during the postharvest life [3]. Their content is affected by genotype, environment and farming practices [4]. Many types of products are obtained from tomato processing: concentrate, peeled, puree, pulp, sauce, dried, and diced. Previous work carried out by De Sio et al. [5] showed that industrial tomato processing changed the content of many compounds, such as increasing soluble solids, reducing sugars, rutin and naringenin, in the fruits of different tomato hybrids. Some quality attributes are closely connected to genotype, such as the shape and size of fruits, content in lycopene, vitamin C and soluble solids. Each of the latter traits has its own specific heritability and is highly dependent on the expression of the genotype in combination with the environment [6]. In particular, lycopene content in tomato fruits reportedly shows a wide variation range, from 6.5 $\mu g \cdot g^{-1}$ to 50.9 $\mu g \cdot g^{-1}$, depending on variety and environment. Similarly, total soluble solid content showed a wide variability, from 2% to 10% [6]. The vitamin C content is also significantly affected by genotype, ranging from 8.9 to 104.6 $\mu g \cdot mL^{-1}$ [6]. The crop cycle length is an important trait in tomato cultivars, positively correlated to fruit size, and indeed the plants that do not have enough time to accumulate biomass cannot support substantial fruit growth [7]. Some chromosomal regions influence tomato earliness, in particular the time span between flowering and ripening, and most of parameters, such as the number of days to flowering, days to fruit ripening, and fruit weight, are affected by the growth habitus [8].

Quality parameters, such as color, dry matter content and viscosity, are crucial for the tomato industry, and show changes after processing whose influence on the final product characteristics depends on the cultivar and the production area [9]. In particular, the dry matter content affects the efficiency of the processing phase, i.e., the percentage of diced produce [10,11], along with peelability, which is also crucial for the product's commercial appreciation [12].

Peelability is one of the crucial fruit features, influencing both their classification and the processing phase, whose efficiency is related to the percentage of fruit integrity upon dicing, and even to dry matter content.

With the aim of producing premium quality products, the main parameters evaluated in both raw and canned produce are dry matter content, titratable acidity, pH, sugars, fatty acids, and citric acid, which contribute to taste and aroma, and are affected by genetic traits and environmental conditions [13]. The epidemiological findings confirmed the beneficial effects on human health of different antioxidants contained in tomato fruits, whose amount and bioavailability can be altered by mechanical and heat treatments as well as the addition of ingredients such as oil or salt, characterizing the industrial transformation into the final products [14]. It was found that the content of antioxidants is significantly affected by the genotype, their content being mostly concentrated inside the skin, followed by pulp, and it is important to investigate on how these compounds vary after the processing phase [15]. Among the antioxidant compounds in tomato fruits, lycopene plays a major role, as a symmetrical, acyclic carotenoid with a peculiar structure characterized by thirteen double bonds, eleven of which are conjugates, which makes it a strong antioxidant [16,17]. Lycopene induces cell-to-cell communications and modulates hormones, immune systems and other metabolic pathways [18]. The lycopene intake through the food is epidemiologically correlated with the reduced risk of prostate cancer and shows a higher inhibition of cell proliferation in various human epithelial cancer cell lines in comparison with α- and β-carotene [19].

The purpose of this work was to assess the effect of processing on the product yield, quality and antioxidant performances of four processed tomato hybrids belonging to the round-prismatic fruit type, grown under organic management in Southern Italy and addressed to the diced industry. Future experiments will be also focused on the growing conditions and the environment, in addition to the comparison between genotypes.

## 2. Materials and Methods

### 2.1. Experimental Protocol and Growth Conditions

This research was carried out in 2018 in Tavoliere delle Puglie (Foggia, southern Italy) on tomato, in order to assess the effects of processing on the product yield and quality of four round-prismatic tomato hybrids addressed to diced production. The experiment was conducted on a silty sand textured soil with 2% organic matter, 1.3 g·kg$^{-1}$ N, 38 mg·kg$^{-1}$ $P_2O_5$ and 95 mg·kg$^{-1}$ $K_2O$. The time course of temperature and rainfall are shown in Figure 1. The tomato hybrids compared were: HMX 4228 (HM.Clause Italia), Max 14111 (Syngenta Italia S.p.A.), UG 16112 (United Genetics Italia S.p.A.), Impact (ISI Sementi).

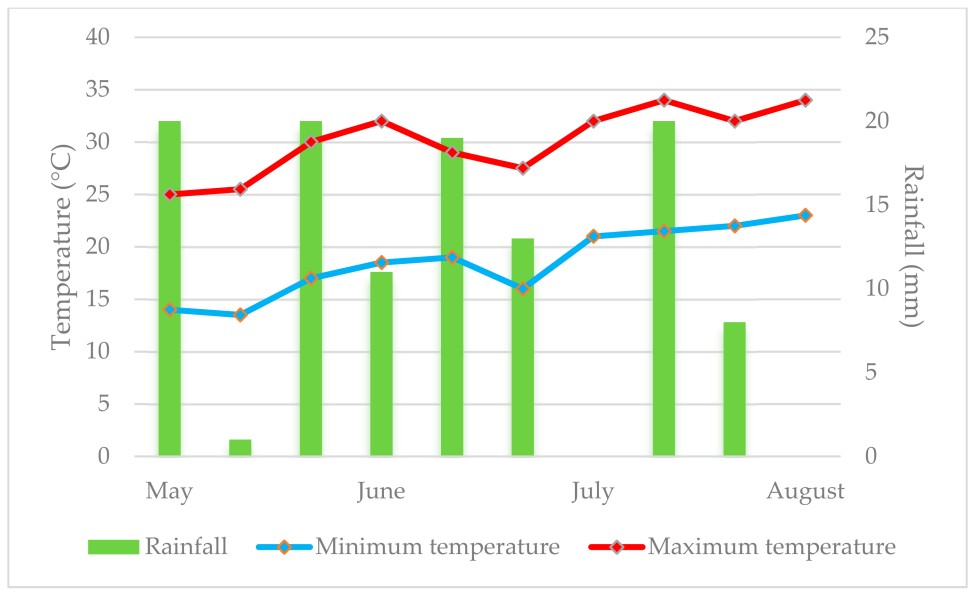

**Figure 1.** Ten-day means of temperatures and rainfall in Tavoliere delle Puglie (Italy) in 2018.

A randomized complete block design with three replicates was applied in the field, tomato hybrid was the only experimental factor and the area of each plot was 65 m$^2$. The previous cultivated crop was wheat, harvested the year before, while tomato was transplanted on 27 April. A double-row layout of plants was arranged, provided with a 12 μm-thick biodegradable mulching cover, and achieving a density of 3.1 plants per m$^2$. The following farming practices were carried out in compliance with EU regulation 834/2007 and subsequent updates. Fertilization with organic and organo-chemical fertilizers permitted for organic system use was practiced by supplying 160 kg·ha$^{-1}$ N, 125 $P_2O_5$, and 112 $K_2O$. The whole phosphorus dose and 50% nitrogen and potassium were applied at planting and the remainder during crop by fertigation: different fertilizers containing N, $P_2O_5$ and $K_2O$ (7% each), N and $P_2O_5$ (6% and 16%, respectively), N and $K_2O$ (7% and 21%, respectively), and N (11%) were used. Twenty drip irrigations, each supplying 60 m$^3$·ha$^{-1}$ water, were carried out over the crop cycles, when the soil available water at 20 cm depth dropped to 80%, based on the crop evapotranspiration [20]. Plant protection against downy mildew, tomato leaf miner, aphids, whitefly and red spider, was managed by using azadirachtin, copper, sulfur and spinosad. The harvests were undertaken manually between 3 and 7 August.

### 2.2. Yield and Processing Efficiency Determinations

The following agronomic determinations were made when 90% of the fruits were ripe: weighing of the marketable fruits (red and color turning point) and waste fruits (green and rotten), of a sample of 100 random fruits, measuring of length and width of 20 fruit samples, calculating the Fruit Shape Index as the ratio of the maximum height length to maximum width of a fruit. Technological, quality and sensory features were determined in SSICA

laboratories (Angri, Salerno). The diced production was carried out on a semi-industrial scale, with the addition of 7.5° Brix juice of the same hybrid, packaged in painted tinplates of 1 kg, sterilizing the product at 100 °C for 90 min in a static bath. Processing yield was assessed through the ratio between canned tomato fruit amount and marketable yield in the field. In the latter respect, each raw tomato fruit sample was divided into two aliquots, one directed to diced processing and the other to juice processing. The waste fractions of both processing chains were determined by weighing yellow, necrotized, rotten, broken, and undersized fruits together with the peel fraction and seeds. The percentage of the product processing yield was calculated as a weighted average of diced and juice processing yield percentages.

### 2.3. Quality Attributes and Antioxidants Determinations

The analytical tests and the consequent determination of the fruit quality components were conducted on both fresh tomato fruit (raw material) and the processed outcome (final product), or only on the final product concerning titratable acidity, proteins, fats, fiber, glucose, fructose, sucrose, fatty acids. In particular, the analyses were performed using the methods reported by: Caruso et al. [21] for total and soluble solids, sugars, titratable acidity, proteins, fats, fiber; Golubkina et al. [22] for fatty acids; Conti et al. [23] for color; De Sio et al. [24] for lycopene; Golubkina et al. [25] for polyphenols. Briefly: sugars were assessed by HPLC, using the 600E Waters chromatographic system and a column Sugar-pak Waters at 85 °C; proteins with Kjeldahl method, by a Foss Tecator digestor with a Kjeltec 2300 distiller; fiber on dried and gelatinized samples enzymatically digested by proteases and amyloglucosidase, with soluble fiber precipitated by ethanol, calculated as the difference to the filtered dry residue weight upon protein and ash determination; sodium by atomic absorption spectrophotometry using a model 1100 Perkin-Elmer spectrophotometer; fatty acids by gas chromatography on capillary glass column, using an Agilent 6890 gas chromatograph equipped with a flame ionization detector; color by a Hunter Associate Laboratories D25-A model colorimeter; lycopene through HPLC, using a Waters Alliance chromatograph equipped with photodiode array detector mod. 996, on a reversed-phase column YMC-Pack C30 (250 mm × 4.6 mm i.d.); polyphenols in water extract through a spectrophotometer (Unico 2804 UV, USA), at 730 nm absorbance, using 0.02% gallic acid as an external standard.

### 2.4. Sensorial Determinations

For each diced tomato hybrid sample, by anonymous coding, a sensorial evaluation test was performed by a panel test of 15 specialists from the tomato industry, composed of 40–60-year-old women and men sitting spatially apart to prevent opinion exchange; each of the aforementioned specialists, provided with water allowing to remove mouth-residual material and taste, evaluated 500 g tomato samples under neutral light (4000 K) over needed time. The experts' opinions were reported in a specific form including 11 sensorial variables, five of primary importance (such as appearance, color, flavor, taste and consistency), and the remainder being detailed with a score scale from 0 (unpleasant) to 10 (pleasant).

### 2.5. Statistical Processing

Concerning the statistical data processing, one-way and two-way analysis of variance (for yield and for quality and antioxidants variables respectively), and Duncan's test of mean separation were applied. Due to non-normality of the data distribution, the angular transformation was applied to percentage values before performing statistical processing: $Y = \arcsin \sqrt{p}$, where p is the original value and Y is the result of the transformation.

## 3. Results and Discussion

### 3.1. Yield and Processing Efficiency

As shown in Figure 1, during the present trial, the fluctuating temperature and rainfall partially hampered the tomato hybrids' potential in fruit setting, also causing a high occurrence of waste fruit. In particular, the air warmed up from 19.6 °C to 24 °C (early May to early June), after which it cooled down to 21.7 °C and heated up again to 27.2 °C (late June to early August). The regular tomato fruit formation and ripeness are closely connected with a proper climatic time course and, indeed, an anomalous meteorological course can influence yield, soluble solids content and flavor, also depending on genotype [26].

As reported in Table 1, tomato hybrids HMX 4228 and MAX 14111 gave the highest marketable yield and the lowest waste fraction in the field, with the latter variable not significantly differing from that associated to Impact. The productive result was the consequence of the highest number of fruits for MAX 14111 and of their mean weight for HMX 4228. The shape index and the flesh thickness were not significantly affected by the hybrid, whereas HMX 4228 showed a higher diameter than MAX and a higher length than Impact. The average yield (76.6 t·ha$^{-1}$ fresh weight) achieved in the present work is consistent with the production values recorded in other investigations on organically grown tomato [27,28].

After the tomato hybrids underwent the industrial transformation, UG16112 showed the worst processing efficiency, with a 13.6% decrease compared to the mean value of the other three hybrids (Table 2). The same trends were recorded for the diced and juice yield, whereas UG16112 had the highest waste percentages both along the peeling and juice chain, which caused the aforementioned lowest yields. The genotype is an important factor influencing the peelability, which is a crucial industrial parameter [29], as well as the red layer and pericarp [30].

### 3.2. Quality Attributes and Antioxidants

In the present research, the industrial processing resulted in higher values of total and soluble solids as well as of reducing sugars, but less intense color, in the tomato diced product compared to the raw fruits (Table 3). As for the comparison between the hybrids, MAX 14111 showed a 9.8% higher total solids and an 8.8% higher soluble solids content compared to HMX 4228, as well as an 8.4% higher level of reducing sugars than the average of Impact and UG 16112. The latter hybrid had the lowest fruit color intensity and HMX 4228 the highest value of sugar ratio. No significant interactions between industrial processing and hybrid on the variables examined were recorded.

Sugars and acids are the main constituents of tomato flavor and affect the quality; total solids and sugars in tomato increase gradually during the fruit growth and ripening [31,32], whereas acidity rises during development and then decreases while ripening progresses [33,34]. A study carried out by Hallmann [35] showed that tomato fruits under organic management were characterized by a significantly higher content of total sugars, organic acids, vitamin C and phenolic compounds such as quercetin-3-O-rutinoside, myricetin and quercetin, compared with conventional farming. Hobson et al. [36] found a correlation between fruit firmness and the colorimetric parameter a; moreover, Gormley and Egan [37], showed a high correlation between compression force and the a/b ratio in tomato fruit.

**Table 1.** Yield and biometrical parameters of processing tomato hybrids.

| | Marketable Fruits | | | | | | | |
|---|---|---|---|---|---|---|---|---|
| **Hybrid** | **Yield** t·ha$^{-1}$ | **Yield** %/Total | **Number Per Plant** | **Mean Weight** g | **Diameter** cm | **Length** cm | **Shape Index** | **Flesh Thickness** mm |
| HMX 4228 | 73.0 ± 1.67 a | 92.4 ± 1.25 a | 25.3 ± 0.62 b | 72.1 ± 1.42 a | 4.8 ± 1.10 a | 5.8 ± 1.17 a | 1.21 ± 0.04 | 7.53 ± 0.26 |
| MAX 14111 | 71.9 ± 1.87 a | 89.9 ± 1.67 a | 28.3 ± 0.82 a | 63.5 ± 1.37 b | 4.4 ± 1.10 b | 5.5 ± 1.10 ab | 1.25 ± 0.05 | 7.50 ± 0.21 |
| UG 16112 | 62.3 ± 1.35 b | 81.0 ± 1.47 b | 22.7 ± 0.85 c | 68.9 ± 1.77 a | 4.6 ± 1.17 ab | 5.5 ± 1.17 ab | 1.18 ± 0.05 | 7.70 ± 0.25 |
| Impact | 62.5 ± 1.78 b | 92.2 ± 1.41 a | 25.3 ± 0.72 b | 61.7 ± 1.18 b | 4.5 ± 1.10 ab | 5.4 ± 1.10 b | 1.16 ± 0.05 | 7.37 ± 0.26 |
| | | | | | | | n.s. | n.s. |

n.s.: not significant; within each column, means ± standard deviations are reported, and the values followed by different letters are statistically different according to Duncan's multiple range test at $\alpha \leq 0.05$.

**Table 2.** Processing yield of four hybrids for diced tomato.

| | Processing Yield | | | Waste Fruits along Peeling Chain | Waste Fruits along Juice Chain |
|---|---|---|---|---|---|
| **Hybrid** | **Product** | **Diced** | **Juice** | **%** | **%** |
| HMX 4228 | 79.3 ± 1.95 a | 60.3 ± 1.57 a | 96.4 ± 2.97 a | 39.7 ± 1.51 b | 3.7 ± 0.17 d |
| MAX 14111 | 78.5 ± 1.28 a | 60.0 ± 1.65 a | 93.5 ± 4.36 a | 40.2 ± 1.81 b | 6.5 ± 0.26 c |
| UG 16112 | 67.6 ± 1.47 b | 51.3 ± 1.32 b | 81.2 ± 4.31 b | 48.8 ± 1.71 a | 18.8 ± 1.37 a |
| Impact | 77.0 ± 1.87 a | 59.7 ± 1.85 a | 90.1 ± 4.15 a | 40.2 ± 1.35 b | 9.9 ± 0.72 b |

n.s.: not significant; within each column, means ± standard deviations are reported, and the values followed by different letters are statistically different according to Duncan's multiple range test at $\alpha \leq 0.05$.

**Table 3.** Quality features of tomato fruits as affected by industrial processing and hybrid.

| Treatment | Total Solids g·100 g$^{-1}$ f.w. | | Soluble Solids °Brix | | Reducing Sugars g·100 g$^{-1}$ f.w. | | Sugar Ratio % | | Color a/b | |
|---|---|---|---|---|---|---|---|---|---|---|
| Industrial processing | | | | | | | | | | |
| Raw | 6.11 ± 0.28 | b | 5.37 ± 0.25 | b | 2.73 ± 0.16 | b | 44.7 ± 2.30 | | 2.55 ± 0.12 | a |
| Diced | 7.53 ± 0.34 | a | 6.54 ± 0.30 | a | 3.38 ± 0.18 | a | 44.9 ± 2.34 | | 1.88 ± 0.10 | b |
| | | | | | | | n.s. | | | |
| Hybrid | | | | | | | | | | |
| HMX 4228 | 6.51 ± 0.79 | b | 5.71 ± 0.67 | b | 3.10 ± 0.38 | ab | 47.7 ± 1.53 | a | 2.26 ± 0.40 | a |
| MAX 14111 | 7.15 ± 0.84 | a | 6.21 ± 0.72 | a | 3.21 ± 0.40 | a | 45.0 ± 1.41 | b | 2.19 ± 0.38 | ab |
| UG 16112 | 6.79 ± 0.82 | ab | 5.90 ± 0.64 | ab | 3.00 ± 0.38 | b | 44.0 ± 1.43 | bc | 2.11 ± 0.38 | b |
| Impact | 6.83 ± 0.76 | ab | 6.00 ± 0.67 | ab | 2.92 ± 0.36 | b | 42.8 ± 1.32 | c | 2.31 ± 0.36 | a |

f.w.: fresh weight; n.s.: not significant; within each column, means ± standard deviations are reported, and the values followed by different letters are statistically different according to Duncan's multiple range test at $\alpha \leq 0.05$.

As observed in Table 4, the fruits of the tomato hybrid MAX 14111 had the highest levels of titratable acidity (not significantly different from UG 16112), of fiber (not different from Impact), and of energetic value. The protein and fat content were not significantly affected by the hybrid.

**Table 4.** Quality features of diced tomato fruits obtained from four hybrids.

| Hybrid | Titratable Acidity g Anhydrous Citric Acid·100 g$^{-1}$ f.w. | Proteins g·100 g$^{-1}$ f.w. | Fats g·100 g$^{-1}$ f.w. | Fiber g·100 g$^{-1}$ f.w. | Energetic Value Kcal·100 g$^{-1}$ f.w. |
|---|---|---|---|---|---|
| HMX 4228 | 0.41 ± 0.017 b | 2.02 ± 0.13 | 0.24 ± 0.04 | 0.86 ± 0.03 b | 26.5 ± 0.62 b |
| MAX 14111 | 0.51 ± 0.020 a | 2.13 ± 0.14 | 0.24 ± 0.05 | 1.12 ± 0.08 a | 28.5 ± 0.75 a |
| UG 16112 | 0.49 ± 0.017 a | 2.03 ± 0.15 | 0.21 ± 0.03 | 0.87 ± 0.09 b | 26.4 ± 0.92 b |
| Impact | 0.40 ± 0.020 b | 2.12 ± 0.14 | 0.22 ± 0.04 | 1.05 ± 0.10 a | 26.1 ± 0.87 b |
|  |  | n.s. | n.s. |  |  |

f.w.: fresh weight; n.s.: not significant; within each column, means ± standard deviations are reported, and the values followed by different letters are statistically different according to Duncan's multiple range test at $\alpha \leq 0.05$.

De Bruyn et al. [38] demonstrated that high sugar and acid contents have a positive effect on taste; moreover, sugars and acids contribute to sweetness and sourness of tomato fruits and appear to be essential in their flavor intensity [39]. The latter parameter is, in turn, correlated with several volatile molecules [38,40,41]. Fruit acidity gives an essential contribution to the flavor of tomato products. Citric acid is the most abundant acid and also the main contributor to the total acidity [34,42]. When the fruit maturity occurs, the acidity decreases due to citric acid loss, and concurrently the pH increases [43]. In addition to the contribution to acidity, other acids, such as glutamic and malic acid, contribute to tomato flavor [44]. The ratio between malic and citric acids has been reported to vary between different tomato cultivars [34,45].

As reported in Table 5, HMX 4228 and MAX 14111 revealed a higher fructose content, whereas UG 16112 and Impact showed a higher sucrose accumulation. A study conducted by Zhao et al. [46], reported a high heritability of the sugars in tomato fruit, whose content also depends on the hybrids; moreover, the concentration of fructose, glucose, sucrose, and galactose is negatively correlated with some morphological traits such as the fruit weight, the equatorial fruit diameter, polar fruit diameter and positively correlated with soluble solid content.

**Table 5.** Sugars and fatty acids in diced tomato fruits produced by four hybrids.

| Hybrid | Sugars | | | Fatty Acids | | |
|---|---|---|---|---|---|---|
|  | Glucose g·100 g$^{-1}$ f.w. | Fructose g·100 g$^{-1}$ f.w. | Sucrose mg·100 g$^{-1}$ f.w. | Saturated g·100 g$^{-1}$ f.w. | Monounsaturated g·100 g$^{-1}$ f.w. | Polyunsaturated g·100 g$^{-1}$ f.w. |
| HMX 4228 | 1.53 ± 0.10 | 1.81 ± 0.07 a | n.d. | 0.07 ± 0.010 a | 0.07 ± 0.010 a | 0.09 ± 0.005 b |
| MAX 14111 | 1.58 ± 0.12 | 1.92 ± 0.07 a | n.d. | 0.07 ± 0.005 a | 0.05 ± 0.010 b | 0.11 ± 0.005 a |
| UG 16112 | 1.48 ± 0.17 | 1.60 ± 0.08 b | 210 ± 20 a | 0.05 ± 0.010 b | 0.06 ± 0.005 ab | 0.10 ± 0.010 ab |
| Impact | 1.46 ± 0.11 | 1.52 ± 0.05 b | 230 ± 20 a | 0.06 ± 0.005 ab | 0.05 ± 0.010 b | 0.10 ± 0.005 ab |
|  | n.s. |  |  |  |  |  |

f.w.: fresh weight; n.d.: not detectable; n.s.: not significant; within each column, means ± standard deviations are reported, and the values followed by different letters are statistically different according to Duncan's multiple range test at $\alpha \leq 0.05$.

With regard to the fatty acid content (Table 5), HMX 4228 and MAX 14111 had higher saturated fatty acids content in the fruits compared with UG 16112; HMX 4228 accumulated the monounsaturated fatty acids at a higher extent than Impact and MAX 14111; the latter hybrid revealed a higher fruit content of polyunsaturated fatty acids in comparison with HMX 4228. Most of the fruit flavor compounds derive from unsaturated fatty acids composition and content. The formation of the flavors from fatty acids occurs after the release of free fatty acids from lipids by lipases and the peroxidation of the specific double bonds of the unsaturated fatty acids [47].

In the present investigation, the total polyphenols as well as the naringenin and rutin content in tomato fruits were higher in the processed than in the raw product (Table 6). With regard to the hybrids, the total polyphenol content was significantly higher in the fruit of MAX 14111 compared to HMX4228. MAX 14111 fruit was the richest in rutin, whereas HMX 4228 fruit showed a significantly higher naringenin content than UG 16112. Lycopene was neither affected by industrial processing nor by hybrid. No significant interactions between industrial processing and hybrid on the variables examined were recorded.

**Table 6.** Antioxidants concentration in tomato fruits as affected by industrial processing and hybrid.

| Treatment | Total Polyphenols mg eq. Gallic Acid 100 g$^{-1}$ f.w. | Rutin mg·kg$^{-1}$ f.w. | Naringenin mg·kg$^{-1}$ f.w. | Lycopene mg·kg$^{-1}$ f.w. |
|---|---|---|---|---|
| **Industrial Processing** | | | | |
| Raw | 35.7 ± 1.5 b | 18.7 ± 2.4 b | n.d. | 158.9 ± 12.8 |
| Diced | 41.0 ± 2.1 a | 49.9 ± 5.8 a | 11.9 | 155.2 ± 15.0 |
| | | | | n.s. |
| **Hybrid** | | | | |
| HMX 4228 | 37.0 ± 2.4 b | 30.6 ± 15.4 c | 6.3 ± 6.9 a | 158.8 ± 15.5 |
| MAX 14111 | 40.3 ± 3.7 a | 39.6 ± 20.0 a | 6.1 ± 6.7 ab | 160.4 ± 16.1 |
| UG 16112 | 38.3 ± 3.7 ab | 31.6 ± 16.0 c | 5.6 ± 6.1 b | 153.7 ± 11.4 |
| Impact | 37.7 ± 2.8 ab | 35.5 ± 17.5 b | 5.9 ± 6.5 ab | 155.3 ± 14.3 |
| | | | | n.s. |

f.w.: fresh weight; n.d.: not detectable; n.s.: not significant; within each column, means ± standard deviations are reported, and the values followed by different letters are statistically different according to Duncan's multiple range test at $\alpha \leq 0.05$.

The polyphenol increase recorded in the diced product in the present research is connected with the effect of thermal processing, which can accelerate the release of phenolic compounds by destroying the cellular components. Though the destruction of the cell membranes could also trigger the release of oxidative and hydrolytic enzymes, degrading antioxidants in fruits, processing temperatures of over 80 °C can denature these oxidising enzymes, thus eliciting the phenolics' release [48].

The increase in rutin and naringenin resulting from the industrial processing is due to increased extractability or de novo synthesis, respectively, rather than to deglycosylation or de-esterification of more complexed flavonoid species [49]. However, contrary to our results Vallverdù-Queralt [50] reported a decrease in naringenin and rutin contents upon the industrial processing of tomato fruits. The variation of rutin and naringenin content from raw to processed fruits may be due to the processing method used, which can cause the antioxidants' degradation, thus resulting in lower content in processed tomatoes [51,52].

Diced tomato is processed like the peeled tomato up to the peeling stage, after which the product is cut into cubes and preheated at 50 °C, and next it is canned upon the sauce addition at 90 °C. The polyphenol content may change from the raw to the canned product due to industrial treatments such as the addition of calcium salt (CaCl$_2$), heating, or freezing, which helps to increase fruit firmness [53].

The results of the present research are consistent with the work of Kaur et al. [54], who found that the average polyphenol content of ten commercial cultivars ranged from 26 to 66 mg GAE·100 g$^{-1}$ fresh weight.

Naringenin and rutin, much like other polyphenols, can contribute to a reduction in allergic reactions as well as increased vascular permeability, bronchial smooth muscle contraction, mucus production, and neutrophil chemotaxis, preventing histamine release [55].

In the present research, no significant differences in lycopene content arose between the hybrids examined, and the average value was slightly higher than those recorded in

previous research focusing on both peeled and diced products [56,57]. In previous studies, lycopene content has been found to increase after processing based on heat or thermal treatments [58,59], or decrease upon sterilization and ultrasonication [60]. The increase in total lycopene content recorded in tomato juice compared to raw fruits by other authors [61] is justified by the enhanced release of this pigment, making it more extractable after heat disruption of the cell membranes and walls, and the disintegration of the cromoplasts, which leads to flavor dissociation between the carotenoids and lipophilic proteins [62,63]. Indeed, lycopene is stable under the conditions of thermal processing [48], though high temperature can result in the decrease in lycopene in heated tomato pulp [64] due to its oxidative destruction [65].

### 3.3. Principal Component Analysis (PCA)

The correlations between the four hybrids compared and the variables examined in diced tomato were processed by PCA analysis (Figure 2). The two principal components shown in the biplot graph overall contributed to 80.1% of the total variability (47.4% and 32.7% for PC1 and PC2, respectively). The relationships between the hybrids and the variables assessed suggest that: MAX 14111 and HMX 4228 are associated with the highest values of field and processing yield as well as reducing sugars and fructose; MAX 14111 showed the highest values of total solids and soluble solids, titratable acidity, fiber and energetic value, and polyphenols; HMX 4228 performed best in terms of sugar ratio, color and naringenin.

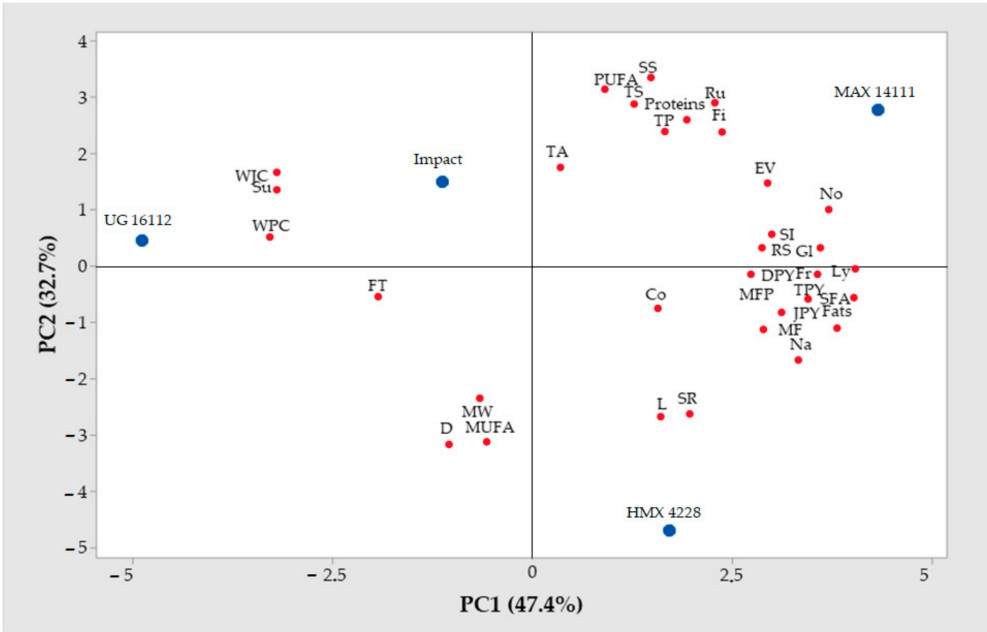

**Figure 2.** Biplot graph relevant to Principal Component Analysis (PCA). HMX 4228, Max 14111, UG 16112 and Impact are the four hybrids tested (blue circles). The variables examined (red circles) are: Co, color; D, fruit diameter; DPY, diced processing yield; EV, energetic value; Fi, fiber; Fr, fructose; FT, flesh thickness; Gl, glucose; JPY, juice processing yield; L, fruit length; Ly, lycopene; MUFA, monounsaturated fatty acids; MW, mean fruit weight; MF, marketable yield; MYP, marketable yield percentage; Na, naringenin; No, number of fruits per plant; PUFA, polyunsaturated fatty acids; RS, reducing sugars; Ru, rutin; SFA, saturated fatty acids; SI, shape index; SR, sugar ratio; SS, soluble solids; Su, sucrose; TA, titratable acidity; TP, total polyphenols; TPY, total processing yield; TS, total solids; WPC, waste along peeling chain; WJC, waste along juice chain.

### 3.4. Sensorial Features

The graphic representation of QDA (Quantitative Descriptive Analysis) obtained by processing the evaluation forms filled in by the experts during the sensorial analysis is shown in Figure 3. In order to make it easier to interpret the high number of data, the sensorial variables of the diced products deemed negative, such as extraneous taste, off flavor and acidity, were extrapolated (Figure 3a), whereas the desired features were displayed in Figure 3b.

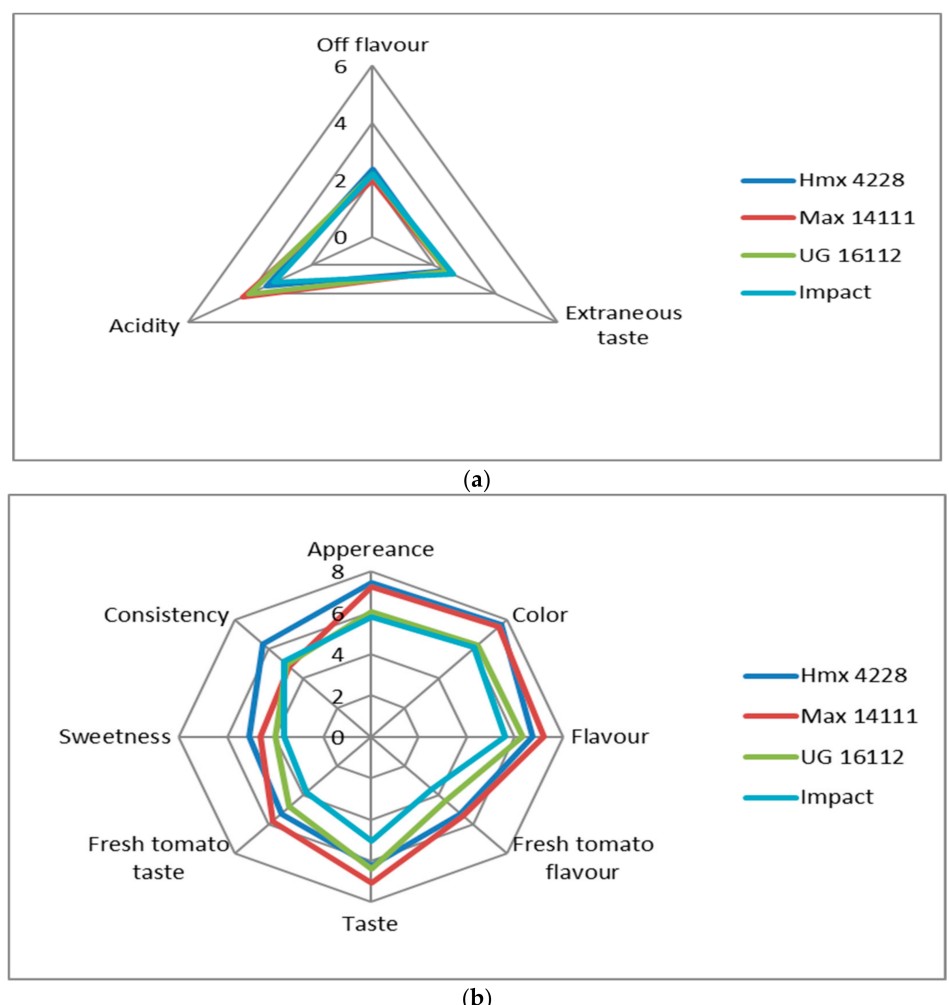

(**a**)

(**b**)

**Figure 3.** Organoleptic evaluations of diced tomato fruits: (**a**) Sensorial profiles of the undesired features, named negative. (**b**) Sensorial profiles of the desired features, named positive.

The sensorial data processed by analysis of variance are reported in Table 7 for the variables which showed statistically significant differences. The aim of the analysis was to determine which groups of means are significantly different from each other. The delta values are the estimated difference between each pair of means. The method used to discriminate between the means is Fisher's least significant difference (LSD) procedure at 95% confidence level. The F crit values represent the Fischer F referred to the aforementioned confidence level and to the degrees of freedom calculated with reference to group size. According to LSD, if delta > F crit the related pairs show statistically significant differences at 95% confidence level. Only the groups showing statistically significant differences have been reported. As observed in Table 7, the cultivars had different performances mainly in appearance and color, whereas no significant differences arose in terms of acidity, consistency, off-flavor and extraneous taste. With regard to flavor, the only difference was observed between Max 14111 and Impact; the latter also differed in sweetness compared

to HMX 4228. The hybrids HMX 4228, Max 14111 and UG 16112 did not show significant differences from each other, as well as Impact and UG 16112.

**Table 7.** Analysis of variance and F significance of the organoleptic features of diced tomato produced by four hybrids, referred to the variables which showed statistically significant differences.

| Sensorial Variables | Cultivar | Cultivar | Δ | F crit |
|---|---|---|---|---|
| **Appearance** | Impact | HMX 4228 | 1.65 | 1.07 |
| | Impact | Max 14111 | 1.45 | 1.10 |
| | HMX 4228 | UG 16112 | 1.4 | 1.10 |
| | Max 14111 | UG 16112 | 1.2 | 1.10 |
| **Color** | Impact | HMX 4228 | 1.6 | 0.97 |
| | Impact | Max 14111 | 1.5 | 0.99 |
| | HMX 4228 | UG 16112 | 1.4 | 1.00 |
| | Max 14111 | UG 16112 | 1.3 | 1.00 |
| **Flavor** | Impact | Max 14111 | 1.6 | 1.27 |
| **Fresh tomato Flavor** | Impact | HMX 4228 | 1.7 | 1.22 |
| | Impact | Max 14111 | 1.8 | 1.25 |
| **Taste** | Impact | Max 14111 | 2.0 | 1.34 |
| | Impact | UG 16112 | 1.3 | 1.27 |
| **Fresh Tomato Taste** | Impact | HMX 4228 | 1.5 | 1.21 |
| | Impact | Max 14111 | 2.0 | 1.24 |
| **Sweetness** | Impact | HMX 4228 | 1.5 | 1.24 |

Δ = differences between the means; F crit expresses the level of significance.

## 4. Conclusions

From the research carried out in Southern Italy on processing tomato, it arose that the diced product derived from industrial transformation, when compared with the raw fruits, showed an increase in some important quality parameters, such as total and soluble solids, reduced sugars and antioxidants (except lycopene, which was not affected by this treatment).

Among the hybrids examined, MAX 14111 and HMX 4228 best fitted the organic management, taking into account that they gave the highest field and processing yield, though the latter was not significantly different from Impact. Moreover, MAX 14111 showed the highest values of most of the quality parameters, such as total solids and soluble solids, titratable acidity, fiber and energetic value, and polyphenols. HMX 4228 also showed the highest levels of some quality parameters such as sugar ratio and color, naringenin, and shared the top ranking with MAX 14111 for reduced sugars and fructose. The obtained results support the choice of sustainable systems for both the field and industrial phase of the tomato supply chain, and prove that the industrial items are a valuable alternative to the fresh ones, thus meeting the increasing consumers' demand for healthy produce and environment.

**Author Contributions:** F.D.S., M.R. and A.D.G. were involved in laboratory analyses; A.C. and G.M. conducted the field experiment and determinations; F.D.S., M.R., A.D.G., A.T. and G.C. contributed to statistically processing and interpreting the data; F.D.S., B.G., L.S. and G.C. conceived and planned the experimental protocol, and performed the research supervision; A.T., N.G., A.S., V.S. and G.C. were involved in bibliographic search; F.D.S., M.R., N.G., A.S., V.S., and G.C. wrote the draft and final manuscript. All authors have read and agreed to the published version of the manuscript.

**Funding:** This research received funding from the seed companies H.M. Clause, Syngenta Italia and United Genetics Italia.

**Institutional Review Board Statement:** Not applicable.

**Informed Consent Statement:** Not applicable.

**Data Availability Statement:** Not applicable.

**Acknowledgments:** The authors wish to thank: Domenico Cacace for his contribution to the laboratory analyses and the statistical data processing; the plant nursery "Aniello Cerrato" in Sarno (Salerno) for producing the seedlings used in the present experimental trial; and the director of Consorzio per la Bonifica della Capitanata (Foggia), Luigi Nardella, for providing the meteorological data related to the research area.

**Conflicts of Interest:** The authors declare no conflict of interest.

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
