# Peer review of "Industrial Processing Affects Product Yield and Quality of Diced Tomato"

_agriculture, doi:10.3390/agriculture11030230_

Round 1
Reviewer 1 Report
Thisstudy was performed with the aim to assess the effects of industrial processing on fruit yield, quality and antioxidants content of four tomato hybrids oriented to diced produce. The topic is interesting if consider that tomato industry has been searching for new genotypes with improved fruit field and processed production as well as quality performances under environmentally sustainable management. The purpose of this work was to assess the effect of processing on the yield, quality and antioxidant performances of four processing tomato hybrids belonging to the round-prismatic fruit type oriented to diced industry, grown under organic management in southern Italy. In addition the authors performed this study in the tavoliere delle puglie, one of the most important area for tomato industry in Italy and in Europe.
I believe that the article is an interesting one. The work plan has been well structured including the analysis carried out. In general, it has been well written and has an adequate bibliography.
Linea121-135: please add major information about chemical methods used; if you want you can insert this info in supplementary material.
Linea 135-138: please improve this section; for example: During the evaluation between one sample and the other what was provided to the judges to rinse their mouths? Quantity of samples.
Tables: please add standard error in all tables.
Table 5: in order to improve the information for reader I suggest to report the sucrose as mg/100 g pf FW.
All tables: Was analysis of variance performed for P-value = 0.05 or for significance level α=0.05 (note applies to all descriptions under the table)?
Author Response
Reviewer 1
Linea121-135: please add major information about chemical methods used; if you want you can insert this info in supplementary material.
Answer: dear Reviewer, we addressed your comments also taking into account that we have been recommended to reduce the similarity percentage in Materials and Methods section (L 344-358).
Linea 135-138: please improve this section; for example: During the evaluation between one sample and the other what was provided to the judges to rinse their mouths? Quantity of samples.
Answer: We addressed the above comments (L 371-376).
Tables: please add standard error in all tables.
Answer: we completed the tables as recommended.
Table 5: in order to improve the information for reader I suggest to report the sucrose as mg/100 g pf FW.
Answer: we did the above mentioned conversion.
All tables: Was analysis of variance performed for P-value = 0.05 or for significance level α=0.05 (note applies to all descriptions under the table)?
Answer: addressed.
Reviewer 2 Report
Agriculture-1047111-v1 Industrial Processing Affects Fruit Yield and Quality.pdf
The submitted manuscript reports the results of an experimental test on tomato cultivation and subsequent industrial processing. An extensive assessment, both on the field production and the industrial quality, with reference to four different tomato hybrids was performed. In particular, productive and quality variables were observed before and after industrial processing, on fresh and diced tomato, respectively, and their differences systematically compared.
The interesting result is that, with respect to several quality traits, the industrial product is higher in organoleptic properties and high-value chemical components than the fresh starting fruits.
A large amount of experimental data were obtained, but they are not exhaustively valorized.
Indeed, the chosen approach considered the treatment of each single experimental variables, one by one, thus producing a systematic but somewhat boring review.
What is notably missing is a general and comprehensive evaluation of all the data set, considering together all the selected variables with reference to the 4 tomato varieties and the two stages of the industrial process, upstream and downstream of tomato processing, respectively.
Indeed, Figure 2 represents an attempt in this direction, which I warmly welcome. Unfortunately, the insertion of Figure 2 appears extraneous to the overall structure of the manuscript. In fact, that type of data processing and representation was neither described in the "materials and methods" section nor commented in the "results and discussion" section. It follows that it represents a non-organic insertion to which the Authors themselves assigned very limited importance.
In order to assign the due value to that huge availability of interesting data, what I recommend is to proceed with a multivariate statistical evaluation. In particular referring, for example, to a "principal component analysis", a "factorial analysis" or also a “cluster analysis” able to discriminate the most influential structure of the experimental variability and how this structure is reflected on the 4 hybrids and the process conditions compared to the field conditions.
The English language used in writing should be significantly improved and some statistical details added to tables and figures. More explicative information should be reported in the description of “materials and methods”.
A decisive flaw in the manuscript, unfortunately, is linked to the main question it intends to address. In fact, there is no information on the reason why some significant quality improvements are observed in processed tomatoes compared to fresh products and which processes or transformations are attributable to these positive changes.
All this considered, the manuscript deserves major revisions and a comprehensive reconsideration.
Reed the annexed file for further specific comments.

Author Response
Reviewer 2
What is notably missing is a general and comprehensive evaluation of all the data set, considering together all the selected variables with reference to the 4 tomato varieties and the two stages of the industrial process, upstream and downstream of tomato processing, respectively.
Indeed, Figure 2 represents an attempt in this direction, which I warmly welcome. Unfortunately, the insertion of Figure 2 appears extraneous to the overall structure of the manuscript. In fact, that type of data processing and representation was neither described in the "materials and methods" section nor commented in the "results and discussion" section. It follows that it represents a non-organic insertion to which the Authors themselves assigned very limited importance.
In order to assign the due value to that huge availability of interesting data, what I recommend is to proceed with a multivariate statistical evaluation. In particular referring, for example, to a "principal component analysis", a "factorial analysis" or also a “cluster analysis” able to discriminate the most influential structure of the experimental variability and how this structure is reflected on the 4 hybrids and the process conditions compared to the field conditions.
Answer: based on the above mentioned recommendations, we performed the PCA including all the variables examined (L 233-276).
The English language used in writing should be significantly improved and some statistical details added to tables and figures. More explicative information should be reported in the description of “materials and methods”.
Answer: the above comments have been addressed as recommended by the Reviewer.
A decisive flaw in the manuscript, unfortunately, is linked to the main question it intends to address. In fact, there is no information on the reason why some significant quality improvements are observed in processed tomatoes compared to fresh products and which processes or transformations are attributable to these positive changes.
Answer: integrative discussion has been added according to the above recommendations (L 197-204; 224-229).
All this considered, the manuscript deserves major revisions and a comprehensive reconsideration.
The following are some specific comments.
Original title:
“Industrial Processing Affects Fruit Yield and Quality of Tomato Hybrids oriented to Diced Produce”
Suggested title revisited:
“Industrial tomato processing affects product yield and its quality compared to the quality of tomato fruit hybrids addressed to diced canning”
Answer: we modified the title as recommended by the Reviewer.
Abstract
23 Please modify according to the following:
“Tomato industry has been searching for new genotypes with improved fruit production, both in the field and industrially processed, together with high-quality performance under sustainable management conditions”.
Answer: addressed (L 25-27).
25-26 Please modify according to the following:
“industrial processing on product yield and quality, such as antioxidants content, of four tomato hybrids addressed to diced canning.
Answer: addressed (L 28-29).
27-28 “gave the highest marketable yield and, along with Impact, the highest processing efficiency”
Answer: addressed (L 29-30).
28 performance and not performances
Answer: addressed (L 31).
32-33 break the line with the new sentence.
Answer: addressed (L 37).
33-34 “The four hybrids differed from each other in appearance and colour”. You are still comparing the four hybrids, therefore I suggest inserting the sentence in line 32, before the beginning of the sentence that compares the raw with the processed tomato. Please, what do you mean with the term “appearance”? Fruit shape, size, or something else? Specify, please.
Answer: addressed (L 34-36).
34-36 Please consider the following sentence arrangement:
“The appreciable fruit yield and quality resulting from both the field (according to the organic farming method of production) and the processing phase represent a promising perspective for identifying improved tomato genotypes addressed to diced canning”.
Answer: addressed (L 38-40).
- Introduction
41-42 Please consider the following sentence arrangement:
Farmers are currently aware that crop systems should be managed achieving high food quality while safeguarding environmental resources.
Answer: addressed (L 46-47).
42-43 Please, delete: “In the latter respect, Organic farming method is deemed to ...”
Answer: addressed (L 47).
44 “as compared to conventional farming”
Answer: addressed (L 49).
45 “In the same way as vegetables directed to the fresh market,…”
Answer: addressed (L 49-50).
46 “organic farming procedures”
Answer: addressed (L 51).
51-53 “The latter molecules represent a healthy resource for humans and a resistance factor to plant adversity also during the postharvest life [3].” Make a new sentence: “Their content is affected by genotype, environment and farming practices [4]”.
Answer: addressed (L 56-57).
56-57 [previous work showed that the content of many compounds] “changes as consequence of the industrial conversion from raw material to processed product, managed with organic farming procedures”. Is the specification “managed with organic farming procedures” necessary? I think it can be deleted.
Answer: addressed (L 61-62).
58 “closely connected to the genotype”
Answer: addressed (L 63).
63 “by the genotype”
Answer: addressed (L 68).
64 “to the fruit size”
Answer: addressed (L 69).
68 “are affected by the growth habit”. Personally, I would prefer “habitus”, but also “habit” is fine.
Answer: addressed (L 73).
71-73 Please, better clarify the sentence and improve the structure. It is hardly understandable and a little bit confusing.
Answer: addressed (L 76-78).
74 “both in raw”. Please, invert: “in both raw”
Answer: addressed (L 82).
77 are you referring to human health? Yes of course, but please, specify. Rearrange the whole sentence to explain better this crucial facet related to the transformation and quality modification in the quality of the final product.
Answer: addressed (L 85-88).
91 “to assess the effect of processing on the product yield”. Please add the term "product", otherwise it can be understood that you are referring to the yield in the field.
Answer: addressed (L 98).
93 “oriented”. As already highlighted, the term “oriented” seems to me not appropriate. You shloulf prefer “directed” or “addressed”.
Answer: addressed (L 100).
- Materials and Methods
96 “in order to assess the effects of processing on product yield and quality of four round-prismatic tomato”
Answer: addressed (L 314).
97 the same as in 93
Answer: addressed (L 315).
97 “The soil used for the experiment was silty-sandy and had 2% …”. Please, I suggest the following rearrangement: “the experiment was conducted on a silty sand textured soil with 2% …”
Answer: addressed (L 315-316).
98 "trend". It is not a trend, but a "time course". A trend, on the other hand, means an increasing or decreasing statistical variation over time.
Answer: addressed (L 316-317).
104 “A randomized complete block with three replicates was applied as experimental field design”. “Tomato hybrid was the only considered experimental factor. The area of each plot was 65 m2”.
Answer: addressed (L 324-325).
105-106 “the previous cultivated crop was wheat, harvested on the year before, while tomato was transplanted on 27 April. A double-row layout of plants was arranged, provided with a mulching cover”. What kind of mulching?
Answer: addressed (L 325-328).
107-111 Considering that the organic farming method was applied, how these cultivations operations were carried out. Obviously avoiding any kind of chemicals, I suppose. Please, give us more insights.
Was the tomato crop systematically watered? How much? How was the irrigation amount and time of irrigation determined?
Answer: addressed (L 329-338).
118-119 “Weighing of yellow, necrotized, rotten, broken, and undersized fruits was performed, together with the skin fraction”. “skin”? Are you referring to tomato peels?
Answer: addressed (L 344-352).
121-123 All the analytical tests and the consequent determination of the fruit quality components were conducted on both fresh tomato fruit (raw material) and the processed outcome (final product), I shall suppose. Please, confirm and specify in the text.
Answer: addressed (L 355-358).
135-136 “a sensorial evaluation test was performed”
Answer: addressed (L 375).
138 What were the considered sensorial traits subjected to the expert evaluation? Are those listed in Table 7? Please, give more information.
Answer: the sensorial traits have been listed in Table 7 and we have also added further information (L 376-381).
138-140 “Concerning the statistical data processing, analysis of variance and Duncan’s test of mean separation were applied. Due to non-normality of the data distribution, the angular transformation was applied to percentage values before performing statistical processing”. Please clarify, what kind of angular transformation? Arcsine, I suppose. Specify.
Answer: addressed (L 385-389).
- Results and Discussion
143 “compromised”. Better “hampered”
Answer: addressed (L 105).
145-147 Is rainfall description so determinant if irrigation was systematically performed? Probably not.
Answer: addressed (L 107).
148 “trend”, see a previous comment (line 98).
Answer: addressed (L 108).
150-151 “Tomato hybrids HMX 4228 and MAX 14111 gave the highest marketable yield (Table 1) and the lowest waste fraction (Table 2)”.
Answer: addressed (L 111-112).
168 Table 2. What is the “total” column related to? Please, give more insights.
Answer: we have replaced ‘total’ with ‘product’ in Table 2, and detailed the relevant information in Materials and Methods (L 347-352) .
178 “ripeness”. Better “ripening”.
Answer: addressed (L 139).
179 “with the ripening progress”. Better “while ripening progresses”
Answer: addressed (L 140).
186 Table 3. Do the quality values of the four hybrids concern fresh fruit or diced product?
Answer: the quality values concern both fresh and diced fruits, as the main effects of the experimental factors ‘industrial processing’ and ‘hybrid’ have been reported in Table 3, as derived by the related two-way ANOVA.
247 Table 6. As for Table 3 (see line 186)
Answer: the previous answer is also worth for this comment.
269 Table 7. Are the data showed on the table obtained by the experts' sensorial evaluation test? Please, better explain the meaning of the reported data and how they are arranged. Is the Delta values in the column equal to the difference between the values of the first and second cultivar columns? What is the meaning of the values reported in the “F crit” column. Is a Least Significant Difference (LSD) or a Honestly Significant Difference (HSD) or a Probability value (%). It can’t be the F-Fisher statistics and it is hard to understand its meaning.
Answer: addressed (L 279-286).
269 According to the text, Figure 2 should be placed after Table 7. Moreover, Figure 2 is not referred to in the text and there are no comments about the three panels of which it is composed. Something is missing and especially a comprehensive characterization of the four tomato hybrids.
Answer: the text and the related Figure (currently Figure 3) have been properly matched in terms of sequence.
Conclusions
276 “expect”. Please use the correct words. “except”
Answer: addressed (L 395).
280 “for” eliminate
Answer: addressed.
283 “The results obtained”. Better: “the obtained results”.
Answer: addressed (L 402).
284 “tomato supply chain” or “tomato value chain”.
Answer: addressed (L 403).
284-286 “prove that the industrial items are a valuable alternative to the fresh ones, thus meeting the increasing consumers’ demand for healthy produce and 285 environment”. Well, it’s still hard to believe (but it is my personal opinion).
Answer: of course we are respectful of your opinion.
Considering all the data Tables, you should insert the standard error (coming from the One-Way ANOVA) and preferably a CV (coefficient of variation) for each considered variable.
Answer: addressed.
Moreover, considering the statistical procedure, why a two-way ANOVA was not performed? Being “hybrids” the first factor, while “fresh fruit vs canned diced” the second one.
Answer: addressed (L 385-386).
Round 2
Reviewer 2 Report
Please, consider the attached file

Author Response
[Agriculture] Manuscript ID: agriculture-1047111 - Review Request
Dear Authors,
Your manuscript was significantly improved and I am satisfied with the changes made to the work that is now more readable, clear and surely attractive. However, the “abstract” still presents some flaws and should be improved. As a general consideration, still minor issues should be solved, but without the need of a further revision from my side.
Here are some specific comments.
TITLE
I apologize to the Authors for my previous suggestion on the title of the article. In this second revision, I realized it was worse (!) than their original proposal. Sorry for this.
Original title:
“Industrial Processing Affects Fruit Yield and Quality of Tomato Hybrids oriented to Diced Produce”
Suggested Title:
“Industrial Processing Affects Product Yield and Quality of Diced Tomato”
Answer: dear Reviewer, we have modified the title in compliance with your recommendation.
ABSTRACT
Line 25-27: “Tomato industry has been searching for new genotypes with improved fruit production, both in the field and industrially processed, together with high-quality performance under sustainable management conditions”. Consider revising
Answer: addressed (L 24-26).
Line 27-29. Consider the following revising: “A research was carried out in Southern Italy with the aim of assessing the effects of industrial processing on yield and quality of four tomato hybrids grown according to the organic farming methods and addressed at dicing”.
Answer: addressed (L 26-28).
Line 29-30. consider the following revising: “MAX 14111 and HMX 4228 gave the highest marketable yield, and along with while Impact the highest processing efficiency.
Answer: we have revised the whole paragraph concerning hybrid yield and quality, starting from this sentence, as recommended in the following comment (L 29-32).
Line 29-34: “Concerning the field quality, ….MAX 14111 and HMX 4228 gave the highest marketable yield, and along with Impact the highest processing efficiency. The best fruit quality performance was shown by: MAX 14111 for total solids and soluble solids; HMX 4228 for sugar ratio and colour; HMX 4228 and MAX 14111 for the reducing sugars and fructose; MAX 14111 for titratable acidity, fiber and energetic value; UG 16112 and Impact for sucrose; total polyphenols in MAX 14111; rutin in MAX 14111 and Impact, naringenin in HMX 4228”.
Consider revising. Could you be more synthetic and effective? You can use the multivariate discrimination performed with PCA statistical procedure or, alternatively, part of the “Conclusion” section. 2
Answer: we have revised the whole paragraph, as recommended by the Reviewer (L 29-32).
Line 34-38. Consider the following revising: “Concerning the diced products, the sensorial qualities of the four hybrids differed significantly”.
Answer: addressed (L 32-33).
Line 38-39. Consider the following revising: “The appreciable fruit yield and quality resulting from both field and processing phase …”
Answer: addressed (L 35).
Line 40: “tomato genotypes addressed at dicing”
Answer: addressed (L 36).
INTRODUCTION
Line 60-62. Consider the following revising: “… showed that tomato industrial processing changed the content of many compounds, such as soluble solids, reducing sugars, rutin and naringenin, in fruits of different tomato hybrids”.
Answer: addressed (L 56-57).
Line 67: Consider the following revising: “Similarly, and total soluble solid content from 2% to 10% showed a wide variability”.
Answer: addressed (L 62-63).
Line 100: “… type, grown under organic management in southern Italy and addressed to diced industry”.
Answer: addressed (L 96).
MATERIAL AND METHODS
Is there a specific reason why this section was now postponed at the end of the manuscript? I don’t think is a standard rule of the journal, but I may be wrong. Or perhaps is a choice by the Authors? For me is ok in any case.
Answer: we have inverted the two sections, as recommended.
Line 379-380. Consider the following revising: “The experts’ opinion was reported in a specific form including 11 sensorial variables, …”
Answer: addressed (L 168).
RESULTS AND DISCUSSION
Line 114: “flesh thickness”
Answer: addressed (L 191).
Lien 114: “shape index”. No information about “shape index” in material and methods. Please, give knowledge about how this index is calculated. Fruit Shape Index is the ratio of the maximum height length to maximum width of a fruit.
Answer: we have added the above mentioned information in Materials and Methods section (L 130-131).
Referring to all the tables in general. Please declare that standard errors (or standard deviations ???) are present in the tables.
Answer: we have reported in the legend of all tables the mentioned statement.
Table 3: Please, in the upper part of the table apply the same mean discrimination standard you have applied in the lower part of the table by using letters “a” and “b”, respectively at the side of each number. Delete the line where you report “*” or “n.s.”
Answer: addressed.
Table 5. Probably you should avoid zeroed average number and simply declare “n.d.” (not detectable), without considering a standard error.
Answer: addressed.
Line 204. Please, delete the reference: (Capanoglu et al., 2008).
Answer: addressed (L 287).
Line 225. “by to”. Please, delete “to”
Answer: addressed (L 308).
Figure 3. What is the reason to represent twice the same variables? Simply report Figure 3b and 3c, without reporting Figure 3a. 3
Answer: we have removed Figure 3a.
Table 7. Still difficult to understand the meaning of “F crit”. The questions raised in the previous revision remain unsolved.
From my previous revision: “Table 7. Are the data showed on the table obtained by the experts' sensorial evaluation test? Please, better explain the meaning of the reported data and how they are arranged. Is the Delta values in the column equal to the difference between the values of the first and second cultivar columns? What is the meaning of the values reported in the “F crit” column. Is a Least Significant Difference (LSD) or a Honestly Significant Difference (HSD) or a Probability value (%). It can’t be the F-Fisher statistics and it is hard to understand its meaning”.
Answer: we have addressed the above mentioned Reviewer’s comments (L 363-376).
CONCLUSIONS
Line 392. Southern (in capitol)
Answer: addressed (L 394).
Line 402. Consider the following revising: “for both”
Answer: addressed (L 404).
